# Human Sinusoidal Subendothelial Cells Regulate Homing and Invasion of Circulating Metastatic Prostate Cancer Cells to Bone Marrow

**DOI:** 10.3390/cancers11060763

**Published:** 2019-05-31

**Authors:** Alessia Funari, Maurizio Alimandi, Luca Pierelli, Valentina Pino, Stefano Gentileschi, Benedetto Sacchetti

**Affiliations:** 1Department of Molecular Medicine, Sapienza University of Rome, 00161 Rome, Italy; alessia.funari@gmail.com; 2Department of Clinical and Molecular Medicine, Sapienza University of Rome, 00161 Rome, Italy; maurizio.alimandi@uniroma1.it; 3Department of Experimental Medicine, Sapienza University of Rome, 00161 Rome, Italy; luca.pierelli@uniroma1.it; 4Università Cattolica del Sacro Cuore, Istituto di Clinica Chirurgica, 00168 Roma, Italy; valentina.pino013@gmail.com (V.P.); stefano.gentileschi@policlinicogemelli.it (S.G.); 5Fondazione Policlinico Universitario “A. Gemelli” IRCCS, Dipartimento Scienze della Salute della Donna e del Bambino, Unità di Chirurgia Plastica, 00168 Roma, Italy; 6Department of Science, University ROMA TRE, 00146 Rome, Italy

**Keywords:** bone marrow (BM), mesenchymal stem cells (MSCs), skeletal stem cells (SSCs), pericytes, CD146, metastatic prostate cancer, humanized bone marrow, in vivo assays

## Abstract

Subendothelial cells (pericytes) are the clonogenic, multipotent and self-renewing skeletal stem cells (SSCs) found in bone marrow (BM) stroma. They express genes maintaining hematopoietic stem cell (HMC) niche identity and, transplanted in immunocompromised mice, organize the hematopoietic microenvironment (HME) generating humanized bone/BM ossicles. To create a mouse model of hematogenous metastasis of human prostate cancer (PC) cells to human bone/BM, we injected PC cells in the blood circulatory system of Severe Combined Immunodeficiency (SCID)/beige mice bearing heterotopic ossicles. Results indicate that PC cells could efficiently home to mice-implanted extraskeletal BM ossicles, but were not able to colonize mice skeletal segments. In humanized bone/BM ossicles, early foci of PC cells occupied a perisinusoidal position, in close contact with perivascular stromal cells. These findings demonstrate the importance of the SSC compartment in recreating a suitable environment to metastatic PC cells. Our data support the hypothesis that BM SSCs committed to a pericyte fate can specify for homing niches of PC cells, suggesting an involvement of specific interactions with subendothelial stromal cells in extravasation of circulating metastatic PC cells to BM.

## 1. Introduction

Bone is a privileged metastatic site for numerous tumors and the most common for prostate cancer (PC) [1]. Various mechanisms have been proposed to explain bone metastasis of PC cells, including the hemodynamic theory [2] and the “seed and soil” theory [3], but identity of PC cell niche in bone marrow (BM) soil remains to be defined. Recent reports have shown that BM specialized vascular structures delineate a microenvironment supporting metastasis of leukemia cells. These small-restricted areas around capillaries and pericytes topically overlap with hematopoietic progenitor cell niche [4] and are characterized by C-X-C motif chemokine 12 (CXCL12) expression [5], whose receptor CXCR4 is frequently overexpressed on primary tumor cells [6]. These areas have been characterized for homing of leukemia cells [6], but they are probably important also for metastasis of circulating solid cancer cells. We have recently demonstrated that Melanoma Cell Adhesion Molecule (MCAM)/CD146^+^ human BM stromal cells (hBMSCs) surrounding BM sinusoids are CXCL12 secreting pericytes, having active roles in establishing of hematopoietic stem cell (HSC) niches [7]. We hypothesized that transplanted hBMSCs would be able to establish a microenvironment suitable to receive and regulate extravasation of injected circulating PC cells to BM.

It is known that human PC cells, including PC-3 cells, are relatively refractory to colonize murine bone [8,9]. PC-3 is a very aggressive human prostate cancer cell line derived from bone metastasis able to produce pure osteolytic lesions when directly introduced into mouse bone, but not to colonize bone or BM when blood-injected [10]. This behavior likely reflects the species-specificity of molecular interactions guiding cell-cell crosstalk, essential for PC and other cancer cell types to stably reside in permissive microenvironments and produce metastasis. 

Engineered “humanized immunodeficient (SCID-hu) mice” implanted with macroscopic fragments of human fetal bone fragments have been successfully utilized as in vivo metastatic models to study bone metastatic potentials of human blood-injected PC cells into transplanted tissues [8,9]. Studies using this model have provided support to the hypothesis of a preferential tropism of human PC cells for human bone, and have even provided limited evidence that cancer cells may home to transplanted tissues when infused in the systemic circulation [8]. Nonetheless, this model leaves a number of requirements unmet. Availability of human fetal tissues may be limited, hampering design and set-up of specific experiments; human skeletal tissues may not remain unchanged once transplanted, but rather trigger tissue reactions including osteoclastic resorption and fibrosis; vascularization of a macroscopic fetal bone fragment may not necessarily be adequate; fetal bone does not reproduce the structural properties, and may not necessarily reproduce the functional properties of adult bone. For these reasons, we have ectopically generated a humanized bone/BM (also defined ossicle) in SCID/beige mice subcutaneous tissues, by transplanting human CD146^+^ BMSCs [7]. In these mice, PC cells are later injected either intra-venously (i.v.) or in the left ventricle (i.c.). We show here that both procedures are followed by colonization and establishment of metastatic foci into mice-generated ectopic bone/marrow ossicles (Figure 1). To our knowledge, this is the only available in vivo model preserving physiology of intra-species epithelial-stromal interactions, prerequisite for the correct evaluation of cell dynamic studies. Moreover, this is the only model where either side of the epithelial-stromal interaction can be directly manipulated to dissect roles of any specific molecular determinants of the metastatic process (e.g., adhesion molecules, matrix proteins, or growth factors). Our results suggest that human CD146^+^ BMSCs, and not bone or bone cells, allow blood-injected human PC cells to resettle, survive and growth into ectopic bone/marrow ossicles (Figure 1A). These results were further confirmed using our second in vivo model, in which PC cells blood-injected in mice carrying heterotopic co-transplants of hBMSCs and HUVECs (Human Umbilical Vein Endothelial Cells) and in the absence of human bone [11], formed metastatic foci (Figure 1B). Our findings demonstrate the importance of an inter-molecular interaction between cancer cells and perivascular hBMSCs as prerequisite for the formation of prostate cancer metastases.

## 2. Results

### 2.1. Homing of Endogenous Hematopoietic Murine Cancer Cells to Human Extraskeletal HME

We generated our SCID-humanized mouse model, by implanting human BM-derived CD146^+^ skeletal stem cells (SSCs) loaded on osteoconductive carriers (e.g., hydroxyapatite/tricalcium phosphate (HA/TCP)) in subcutaneous tissues of immunocompromised SCID/beige mice (Figure 2A). After eight weeks, implanted SSCs were able to differentiate and recreate a BM hematopoietic microenvironment composed by human-derived skeletogenic tissues (e.g., bone, cartilage, fat and perivascular stromal cells), and mouse-derived hematopoietic cell lineage [7]. Although with low frequency (~15%), immunocompromised mice are known to develop T cell malignancies over time [12], and this occurred also in our SCID/beige mice bearing heterotopic ossicles. In this in vivo experimental model, endogenous leukemic T cells were monitored for their ability to produce metastasis in mouse and/or in human bone/BM ossicle. Complete mice necropsy revealed widespread dissemination of leukemic cells in canonical sites such as liver, spleen, and BM, but forming metastatic foci also in heterotopic bone/BM ossicles (Figure 2B(a–f)). Analogously with normal BM hematopoiesis, hematopoietic tissues normally colonizing heterotopic humanized bone/BM ossicles were displaced by a monotonous population of CD3-expressing lymphoblastic cells. (Figure 2B(e)). As expected, human adventitial reticular cells expressing CD146 in extraskeletal bone/BM ossicles were detectable around sinusoids, both in mice free of associated malignancies [7] and in lymphoma-bearing extraskeletal bone/BM ossicles (Figure 2B(f)). These results indicated the presence of an intact sinusoidal/perisinusoidal microenvironment preserving cell dynamic interactions, even in the presence of hematopoietic malignant cells. Identical experiments were conducted in SCID/beige mice where extraskeletal bone/BM ossicles were generated by transplanting human BM SSCs cultured in vitro to form cartilage pellets (Figure 2C). Also in this case, circulating leukemic cells were able to form metastases both in host murine BM and in extraskeletal humanized bone/BM ossicles. 

### 2.2. Human PC Cells Do Not Efficiently Colonize Adult Mouse Bone/BM

It has been reported that either intra-venously (i.v.) or intra-cardiac (i.c.) injected human PC cells are refractory to colonize mouse bone/BM [8,9,13], an attitude that contrasts with the elected tropism that PC cells normally have for human bone and BM. These considerations brought us to design an experimental in vivo model to evaluate metastatic steps of prostatic adenocarcinoma PC-3 cells injected in the circulation of mice bearing extraskeletal bone/BM ossicles (Figure 3A). We first assessed CD146 expression in human PC-3 cells (Figure 3B(a,b)) stably transduced with Green Fluorescent Protein (GFP) (PC-3-GFP) (Figure 3C) before injection in SCID/beige mice. As expected, PC-3-GFP cells express high levels of CD146, as detected by Western Blot (WB) and Fluorescent Activated Cell Sorting (FACS) analysis (Figure 3B,C). In our experiments, we first used i.c. injection to deliver GFP-tagged PC-3 cells in SCID/beige mice with the intent to monitor their metastatic abilities in mouse bone/BM. We found that GFP-tagged PC-3 cells were not able to colonize BM (Figure 3D) but could only transiently be detected in mouse BM at early time points post-injection (three days) (Figure 3D). Five weeks post-injection, histological and FACS analyses did not reveal PC-3 cells in mouse BM (Figure 3D), but the presence of a morphologically intact marrow organization. Furthermore, no signs of metastatic deposits or radiographically detectable lytic lesions were observed in bone/BM mice (Figure 3E).

### 2.3. Efficient Homing of Human PC Cells to Perisinusoidal Niches of Mice-Implanted Heterotopic Bone/BM

Experiments were then repeated in immunocompromised mice hosting extraskeletal bone/BM generated by human SSCs transplantation. About eight weeks from subcutaneous transplantation of human SSCs with HA/TCP, we injected 10^6^ PC-3 cells directly in the left ventricles of mice. In alternative experiments, we use mice tail veins to inject PC cells (Figure 4A). About three weeks post-injection, mice were sacrificed, and subjected to high resolution radiography and complete necropsy. No evidence of tumor colonization could be detected in murine skeleton, neither radiographically (Figure 4B) or histologically. In control transplants, we noticed development of heterotopic hematopoietic foci around the newly established sinusoids, most of them surrounded by human CD146-expressing stromal cells. By contrast, ~50% of extraskeletal bone/BM ossicles harvested from mice that received blood-injected PC-3 cells were colonized by metastatic foci (Figure 4C(a–f) and Table 1). This is consistent with previous observation indicating that circulating human PC cells were not able to colonize mouse bone, but could relocate and grow within the heterotopically human microenvironment created by transplanted SSCs. (Figure 4C(a–f)) Refined histological analyses conducted in four week old extraskeletal implants revealed the presence of abundant bone and fibrous tissue, but not BM (Figure 5a,b); a population of human CD146^+^ cells were mixed to fibrous cells and could be systematically detected in contact with vessels. About eight weeks post-transplantation, a complete heterotopic BM microenvironment had been recreated in ossicles, and CD146-expressing cells were observed as perivascular stromal cells around BM sinusoids (Figure 5c,d). About eight weeks post-transplantation of human CD146^+^ BMSCs, PC-3 cells were systemically blood-injected in SCID/beige in mice bearing heterotopic ossicles. In this case, cancer cells were able to seeded and grown in ectopic BM ossicles to form metastatic cancer foci (Figure 5e,f). Like in human BM (Appendix A) and in heterotopic bone/BM ossicles, sinusoids connected with early metastatic deposits were coated with an adventitial layer of human CD146^+^ stromal cells. Here, low-expressing CD146^+^ stromal cells were physically associated with sinusoids and detectable as a tumor-associated stroma (Figure 5g,h). Single human PC-3 cells could be frequently detected within the lumen, or adhering to the luminal surface of endothelial cells, or surroundings sinusoids (Figure 6a–c). Small solid clusters of PC-3 cells, representing local growth of tumor cells, identified by CD146 in heterotopic BM microenvironment were likewise associated with sinusoids (Figure 6d–i). These data indicate the presence in extraskeletal BM ossicles of “sinusoidal niches”, not dissimilar from BM hematopoietic progenitors niches functioning as a privileged site for homing of PC cells. 

### 2.4. CD146^+^ hBMSCs Are Sufficient to Establish Human PC-3 Cells Extravasation

Since our previous experiments demonstrated that cancer cells do not establish functional interaction with bone or bone matrix during the metastatic processes, we hypothesized that epithelial cancer cells would interact with vascular cell compartment to form metastases. We designed an experimental model to verify if the metastatic processes from cell extravasation to humanized BM niche colonization would depend on species-specific interaction between human BM stromal cells and human cancer cells. Tagged human BMSCs suspended in a non-osteoconductive growth factor-reduced (GFr) Matrigel^TM^ carrier were locally transplanted subcutaneously in the back of the immunocompromised SCID/beige mice (Figure 7A). Control experiments were done in mice utilizing Matrigel^TM^ grafts of human foreskin fibroblasts. We harvested Matrigel plugs from the first groups of mice, respectively at three or four weeks from implantation. No bone structure or vessel formation were observed at both times (Figure 7B(a–d)) and circulating heart-injected human PC-3 cells were unable to colonize the hBMSCs/Matrigel plugs. Matrigel^TM^ GFr did not negatively influence human PC-3 cells growth capacity. Indeed, when PC-3 cells were suspended in GFr Matrigel^TM^ matrix and directly subcutaneously engrafted in SCID mice, PC cells grew efficiently forming tumor foci vascularized by murine vessels (Figure 7B(e)). By contrast, when hBMSCs and HUVECs were subcutaneously co-transplanted in Matrigel^TM^ scaffolds, in SCID/beige mice, there was no formation of bone structures, but only a system of functional blood vessels, formed by HUVECs hCD34^+^ endothelial cells and hBMSCs-derived pericytes (Figure 7B(f)). These vessels were perfectly functional and they were able to bear heart-injected human PC-3 cells and mediate the formation of metastatic foci of PC-3 in 2/15 Matrigel plugs. Indeed, using this model of hematogenous cell dissemination (Figure 8A), we injected PC-3 cells in the vein tail of SCID/beige mice carrying Matrigel^TM^ matrix with hBMSCs and HUVEC cells co-transplanted (Figure 8B(a)). Following this procedure, hBMSCs were able to generate CD146^+^ perivascular cells around blood vessels and PC cells were able to colonize the three week old implants (Figure 8B(b,c)). In this case, injected PC-3 cells were able to grow, creating metastatic foci in close contact with SSC BM perivascular cells (Figure 8B(b,c)), and forming extensive metastases in about eight weeks (Figure 8C).

## 3. Discussion

Bone and BM are good soils for tumor growth, but some tumors and in particular prostate carcinoma are attracted here through specific cell–cell interactions. Evidence suggests that MSCs increase the cancer cell metastatic potency and tumor growth, immune evasion, and resistance to chemotherapy [14,15,16,17,18,19,20]. In particular, interactions between tumor cells and BMSCs play a major role in supporting prostate cancer growth and survival in bone [21,22,23]. Expression of CD146 has been correlated with metastatic abilities of several tumors, including melanoma and prostate cancer [17,24,25]. The mutual expression of CD146 in metastatic cancer cells and perycites surrounding BM capillaries suggests an involvement of this cell adhesion molecule in tumor angiogenesis and metastasis [17]. Using our new animal models, we observed that human PC cells are specifically attracted by the BM microenvironment, particularly by human CD146^+^ BM pericytes, even in bone absence. SSCs are MSCs, perivascular stromal cells in the BM [7], and progenitors of all tissues that together comprise the bone–BM organ (bone, cartilage, fat and perivascular stromal cells); SSCs can be prospectively isolated based on phenotype. They generate clonal progenies in vitro and replicate the development of bone/marrow organ, including bone and BM, in defined experimental transplantation systems [7,11,26]. Indeed, SSCs derived from human BM and CD146-expressing are MSCs able to transfer the HME and to establish BM niche identity in HME, upon subcutaneous extraskeletal transplantation in immunocompromised mice [7]. In BM HME niches, molecular processes regulating survival, proliferation and HSC differentiation significantly overlap with tumor-initiating mechanisms actuated by cancer cells during metastatic processes. Therefore, it is not surprising that certain cancer cell types show a tendency to form metastatic foci in BM, where they may lodge in the pre-existing supportive stromal microenvironment. These mechanisms have been modeled and clarified for leukemia cells [4,5], while the same concept applied for solid cancer field is relatively new. Indeed, the status of a “permissive and supportive microenvironment” is now becoming an attractive model for the comprehension of molecular mechanisms regulating metastasis of solid tumors. 

One question is how and why epithelial cancer cells colonize and metastasize to the BM. A lot has been written about homing of MSCs to tumors, but little is known about homing of solid tumor cells to the natural niche of MSCs, the BM niche. Most information derives from studies where human MSCs are blood-injected in mice; human MSCs are then monitored for the abilities to reach tumor sites, where they often exert a tumor promoting activity. Our work starts from the opposite point of view: the injection of cancer cells and their homing to MSC BM niches or in a heterotopic site where human BM has been recreated by MSC transplantation. To study the interaction of metastatic cancer cells with human BM niches, we utilized a new model developed in our laboratory to study the interaction of metastatic cancer cells with human BM niches. Human BM niches were recreated in vivo*,* generating a humanized ossicle including bone and BM, obtained by subcutaneous transplantation of human BM MSCs in immunocompromised (SCID/beige) mice [7,11,26]. This model, while respecting species-specificity of cell–cell interactions, utilizes mice as recipients, and appears appropriate to answer the question: what makes bone an attractive metastatic site for some tumors and in particular for prostate cancer cells? Indeed, our work demonstrates that circulating human PC cells are unable to home and colonize murine BM, but efficiently reach and stabilize in human heterotopic humanized bone/marrow ossicle. Transplantation of human CD146^+^ stromal progenitors establishes a yet to be identified species-specific molecular interaction required for PC cells homing to BM ossicle. Cells providing these cues are unknown. These interactions with metastatic cancer cells can in principle be functional mediated by endothelial cells, hematopoietic cells, osteoclasts, osteoblasts/osteocytes, bone matrix, or BM stromal cells. Among these, only bone matrix, osteoblasts/osteocytes, and stromal cells are human in the recreated extraskeletal ossicle. All of the other cellular components are murine as in the non-permissive murine BM environment. These observations strongly support the hypothesis that the properties of the microenvironment facilitating homing of cancer are specifically associated with one specific human cell type in the recreated BM sinusoids. It appears instead that functional properties and molecular cues of heterotypic ossicle facilitating homing of cancer cells are equally valid for homing of circulating hematopoietic progenitor cells in BM, and seem to be mediated by skeletal progenitor/stem cells surrounding sinusoids. Transplantation of human stromal progenitors defines a suitable environment (niches) for host (murine) hematopoietic cancer cells, as it occurs for normal murine hematopoietic progenitors. Human stromal progenitors define a suitable environment for metastasis of blood-borne, human, non-hematopoietic cancer cells. In our model, cell composition of extraskeletal ossicle reaches mature tissue organization about eight weeks post-transplantation; at this developmental stage, ossicle architecture is complete, being constituted by all cell types forming BM, bone marrow stroma, and sinusoids and is permissive to colonization of injected epithelial PC cells. By contrast, four-week post-transplantation when ossicles are formed only by primitive bone structures and osteoblasts [7], circulating PC cells are unable to settle and form metastatic foci. Hence, osteoblasts/osteocytes or bone matrix do not provide the critical cues for cancer cells homing to bone. These cues can be mediated by the stromal progenitor cells, providing the functional molecular interactions with PC [23,24,27,28]. Stromal progenitors reside over sinusoids. Analysis of nascent metastasis in heterotopic BM ossicle shows that single cancer cells could be specifically found in a peri-sinusoidal space, most of which were coated with transplanted human CD146-expressing stromal cells. Moreover, we found that further growth of PC cells that have homed to the BM does not require the presence of bone *per se*. Since further growth does not depend on bone matrix-derived cues, homing to BM environment is a critical event of bone metastasis, depending on interactions between cancer cells and stromal progenitor cells. What we call a bone metastasis is in fact metastasis to the BM stroma, or to the BM hematopoietic “niche”. These data indicate that a sinusoidal niche, possibly related to the one sought by blood-borne hematopoietic progenitors in tumor free extraskeletal BM ossicle, was the prime site of homing of circulating PC cells.

## 4. Materials and Methods

### 4.1. Cell Cultures and Reagents

Human BM stromal cells (hBMSCs) were isolated and cultured as per established methods [7,29] from BM aspirates. Human subjects provided us with an oral assurance of their willingness to participate in research studies on human tissues was approved by the Research Ethics Committee of Istituto Superiore di Sanità of Rome (approval date 20 September 2016; Prot. PRE-686/16). Human foreskin fibroblasts (ATCC, CRL-2429), human prostate carcinoma cell lines PC-3 (ATCC, CRL-1435) and RWPE-1 (CRL-11609) were obtained from American Tissue Culture Collection and cultured according to ATCC protocols. Human Umbilical Vein Endothelial cells (HUVECs; Cambrex Corporation, East Rutherford, NJ, USA) were grown in Clonetics EGM-2 medium (Cambrex Corporation, East Rutherford, NJ, USA), as per the manufacturer’s instructions. All other tissue culture reagents were supplied by Gibco (Invitrogen, Rome, Italy) if not otherwise indicated.

### 4.2. FACS Analysis

Human prostate carcinoma cell lines PC-3 were incubated with PE-conjugated anti-CD146 antibody (P1H12, BD Biosciences Labware, San Diego, CA, USA) and expression of marker was assessed using a FACSCalibur flow cytometer and Cell-Quest Pro software (version 6.1, Becton Dickinson Biosciences, San Diego, CA, USA).

### 4.3. Immunodeficient Mice

In vivo experiments were performed in 8- to 10-week-old severe combined immunodeficiency (SCID) beige mice (CB17/Icr.Cg-Prkdc^SCID^Lyst^bg^/Crl) from Charles River Laboratories (Wilmington, MA, USA). All animal procedures were approved by the relevant institutional committees (DM n. 98/2011-A, Italian Minister of Health). 

### 4.4. Lenti-Viral Vectors 

Lentiviral vectors for green fluorescent protein (EGFP) expression were produced and used as described [30]. In some experiments, PC-3 cells were transduced with GFP-lentiviral vectors.

### 4.5. In Vivo Transplantation and In Vivo Experimental Bone Metastases

In vivo transplantation of hBMSCs on hydroxyapatite/tricalcium phosphate (HA/TCP) carrier was performed as reported [7,29,31] and implants were subsequently harvested at different time points. Briefly, operations were performed under sterile conditions under anesthesia achieved by intramuscular injection of a mixture of Zoletil 20 (Virbac; 5 mL/g of body weight) together with Rompun (Bayer, Leverkusen, Germany; 1 mL/Zoletil 20 bottles). The mouse back was disinfected with betadine and mid-longitudinal skin incisions of about 1 cm in length were made on the dorsal surface of each mouse. Subcutaneous pockets were formed by blunt dissection. A single transplant was placed into each pocket with up to four transplants per animal. The surgical incisions were closed with surgical staples. In parallel experiments, hBMSCs were grown for 21 days in 15 mL polypropylene conical tubes at a density of 300,000 cells/tube in chondrogenic differentiation medium (Dulbecco’s modified Eagle medium-High glucose (Euroclone, Milan, Italy,) supplemented with ITS^TM^ premix (5 μg/mL insulin, 5 μg/mL transferrin, 5 ng/mL selenious acid, BD Biosciences Labware, San Diego, CA, USA), 1 mM pyruvate (Sigma-Aldrich, St. Louis, MO, USA), 50 μg/mL 2-phosphate-ascorbic acid (Sigma-Aldrich, St. Louis, MO, USA), 100 nmol/L dexamethasone (Sigma-Aldrich St. Louis, MO, USA), and 10 ng/mL TGFβ-1 (R&D systems, Minneapolis, MN, USA). After 21 days of in vitro culture, cartilaginous pellets were transplanted subcutaneously into 8- to 10-week-old anesthetised SCID/beige mice (CB17/Icr.Cg-Prkdc^SCID^Lyst^bg^/Crl, Charles River Laboratories, Calco, Italy) as previously described [26]. Animals were sacrificed at different time points and implants harvested for histological analyses.

In vivo matrigel implants were generated by suspending 1 × 10^6^ hBMSCs in 1ml of Matrigel^TM^ Growth factor-reduced (BD Biosciences Labware, San Diego, CA, USA), either alone or with an equal quantity of HUVEC (Cambrex Corporation, Walkersville, NJ, USA). Aliquots of about 0.7 mL were injected subcutaneously in the back of the immunocompromised mice, carefully positioning the needle between the subcutaneous tissue and the muscle layer (epifascial space). Xenografts were generated by suspending 0.1 × 10^6^ PC-3 cells in 0.4 mL of Matrigel^TM^ Growth factor-reduced, either alone or with hBMSCs [14,32]; cells were injected subcutaneously in the back of the immunocompromised mice. For intra-cardiac injection, at the established times, tumor cells were harvested by trypsinization, passed through a cell strainer (Becton Dickinson, Lincoln Park, NJ, USA) to remove cellular aggregates and washed in PBS. 10^6^ cells (PC-3, RWPE-1) in 0.1 mL PBS were slowly inoculated in the left ventricle of the heart or in the tail vein of each mouse through a 29½ gauge needle on an insulin syringe. To evaluate osteolytic lesions, mice were subjected to anaesthesia, laid down in a prone position, and exposed to X-rays at 25 Kv for 8 s in a Faxitron MX-20 X-ray machine (Faxitron X-Ray Corp., Buffalo Groove, IL, USA) with Kodak MIN-R mammography film (Carestream Health, Rochester, NY, USA). At fixed time points, all animals were sacrificed and subjected to necropsy to harvest the implants, along with the heart, liver, spleen, lungs, forelimbs and hind limbs, in order to evaluate the presence of metastases. 

### 4.6. Recovery of Tumor Cells from Murine BM

In addition, 10^6^ tumour cells were injected in the heart of each SCID/beige mouse as described above and mice sacrificed either at three days, three weeks or five weeks post-injection. The entire marrow content of at least three long bones (femur, tibia and humerus) was harvested by flushing with phosphate buffer without calcium and magnesium. Single cell suspension was prepared by repeated passages through syringe needles of decreasing gauge. After red blood cell lysis, cells were FACS analyzed to detect GFP-positive cells. The results were further confirmed by FACS analysis using specific human marker of the single cell line, like MCAM/CD146 for PC-3 cells (from BD Pharmingen, San Diego, CA, USA).

### 4.7. Immunohistochemistry and Immunolocalization Studies

Heterotopic transplants of humanized extraskeletal bone/marrow organ and matrigel implants, harvested from mice, were snap-frozen in Optimal Cutting Temperature (OCT) embedding medium in liquid nitrogen and cryostat sectioned serially, or alternatively fixed in 4% formaldehyde in phosphate buffer, (decalcified in 10% EDTA for humanized extraskeletal BM only), and processed for routine paraffin embedding. Five-μm thick paraffin sections were used for histology staining with hematoxylin/eosin (H&E), immunofluorescence and immunoperoxidase studies. All primary antibodies used for immunolocalization studies are listed in Table 2 and were used as per standard immunoperoxidase 3,3’-diaminobenzidine (DAB) reaction counterstained with hematoxylin) or immunofluorescence protocols. Secondary antibodies labeled with Alexa Fluor 594 and 488 were purchased from Molecular Probes (Invitrogen, Carlsbad, CA, USA). Fluorescence images-stacks were obtained using confocal microscopy laser scanning (Leica TCS SP5, Leica microsystems, Mannheim, Germany). Brightfield light microscopy images were obtained using Zeiss Axiophot microscope (Carl Zeiss, Germany). 

### 4.8. CD146 Expression in Tumor Cells

Proteins were extracted as previously described [7,29], separated on NuPAGE Novex 4–12% Bis-Tris gels (Invitrogen) under reducing conditions, and transferred onto PVDF membrane (Invitrogen). Immunoblotting was performed with goat polyclonal anti-CD146 1:2500 and anti-actin 1:15,000 (both from Santa Cruz Biotechnology, Santa Cruz, CA, USA).

### 4.9. Statistical Analysis

Statistical analysis was performed by one-way ANOVA and subsequently by Bonferroni post-tests. Differences are considered statistically significant at *p* ≤ 0.05.

## 5. Conclusions

Properties of the microenvironment facilitating cancer cell homing are specifically associated with one specific cell type in BM sinusoids. The same cell type has the property of establishing a hematopoietic microenvironment for circulating hematopoietic progenitors, and is itself a skeletal progenitor/stem cell [7,11]. Transplantation of human stromal progenitors defines a suitable environment (niche) for host (murine) hematopoietic cancer cells, as it does for normal murine hematopoietic progenitors. They also define the environment for metastasis of blood-borne, human, non-hematopoietic cancer cells. 

Analysis of nascent metastasis in heterotopic ossicles shows that single prostate cancer cells home to perisinusoidal space and grow into initial perisinusoidal tumor foci where CD146 expressing stromal cells reside. CD146 expression has been correlated to BM metastatic ability of several tumors, including melanoma and prostate cancer [24,27]. The observation that CD146 is not only expressed by several metastatic cancer cells but also by pericytes around BM capillaries suggests an involvement of this cell adhesion molecule in tumor angiogenesis and metastasis.

## Figures and Tables

**Figure 1 cancers-11-00763-f001:**
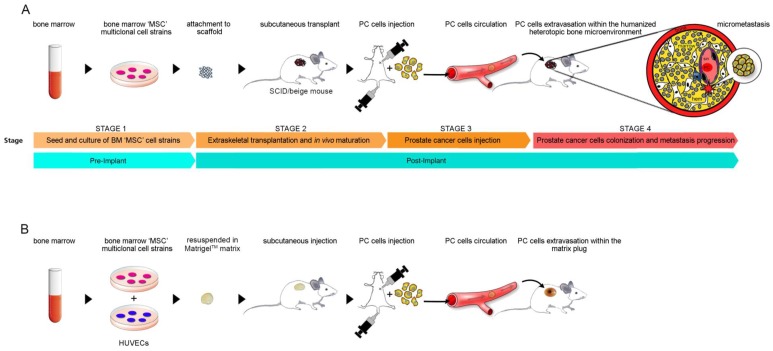
(**A**) experimental design for generation of prostate cancer metastasis in Severe Combined Immunodeficiency (SCID)/beige mice bearing heterotopic humanized bone/Bone Marrow (BM) ossicles; (**B**) experimental design for generation of prostate cancer metastasis in SCID/beige mice carrying heterotopic Matrigel^TM^ matrix with co-transplanted hBMSCs and HUVECs. Prostate cancer (PC) cells. Human Umbilical Vein Endothelial cells, HUVECs.

**Figure 2 cancers-11-00763-f002:**
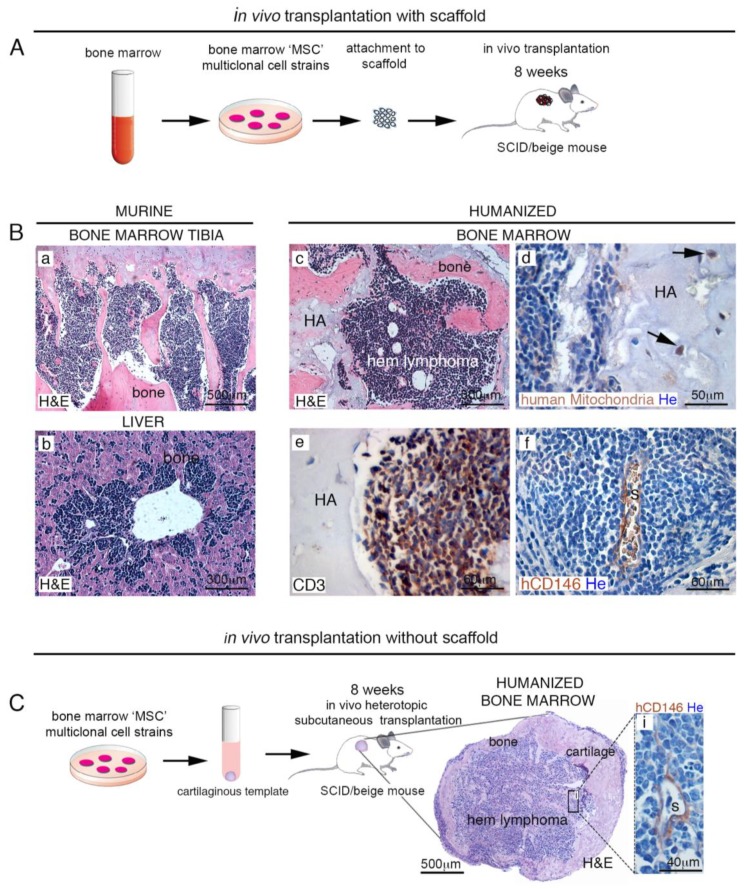
Mouse models for generation of humanized extraskeletal bone/BM ossicles. (**A**) scheme of in vivo transplantation protocol with scaffold. In this model, ectopic bone/BM organ is generated by hBMSCs expanded in vitro, loaded onto osteoconductive carriers (e.g., using hydroxyapatite/tricalcium phosphate (HA/TCP)) and then transplanted in immunocompromised mice. About eight weeks post-transplantation, hBMSCs form a BM microenvironment. (**B**) In the experimental time-frame, a small number of SCID mice (~15%) developed spontaneous lymphomas. (a,b). Human hematopoietic supporting stromal cells in humanized bone/BM ossicles create a suitable environment for murine lymphomas metastases (c–f); (**C**) scheme of in vivo transplantation protocol without scaffold. In this model, cells are grown in culture as unmineralized pellets in chondrogenic differentiation medium and then transplanted into subcutaneous tissues of immunocompromised SCID/beige mice to generate heterotopic skeletal tissues. Eight weeks post-transplantation, hBMSCs develop a BM murine hematopoietic lymphoma-microenvironment. H&E, Hematoxylin/Eosin. HA, hydroxyapatite; hem, hematopoiesis; s, sinusoid. He, Hematoxylin counterstaining. All data shown are representative results derived from one of at least three independent experiments. Scale bars represent 40, 50, 60, 300, 500 μm as indicated.

**Figure 3 cancers-11-00763-f003:**
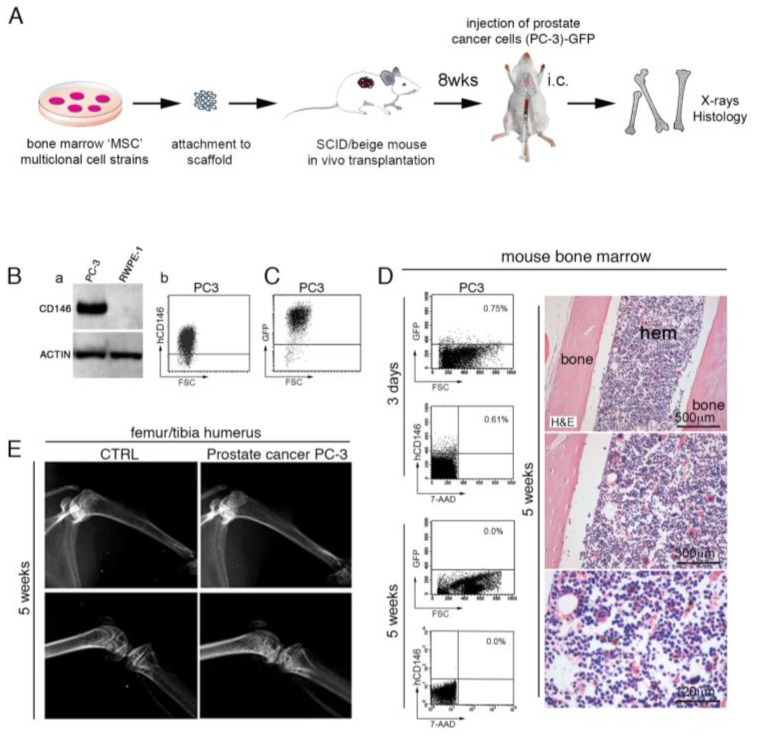
(**A**) human PC-3 and RWPE-1 cells were intracardiac (i.c.) or intra-tail vein (i.v.) injected in mice bearing heterotopic humanized bone/BM ossicles. (**B**) Western blot analysis of PC3 and RWPE-1 cell lysates. β-actin was used as control (a). Fluorescent Activated Cell Sorting (FACS) analysis of PC3 cells stained with anti-human CD146 (b). (**C**) GFP expression of PC-3 cells performed before i.c. injection of SCID/beige mice. (**D**) I.c. injected PC3 cells not survive and growth in murine BM. Representative FACS analysis for human CD146 and GFP in murine BM three days and five weeks after i.c. injection of 1 × 10^6^ PC3 cells show that human PC3 cells reach murine BM, but do not survive or generate histologically detectable metastatic deposits. PC-3 cells are refractory to colonize murine bone/BM. Scale bars represent 120, 300, 500 μm as indicated. (**E**) X-ray analysis did not detect any bone metastasis in mouse skeleton following prostate cancer PC3 or RPWE-1 injection. H&E, Hematoxylin/Eosin; hem, hematopoiesis.

**Figure 4 cancers-11-00763-f004:**
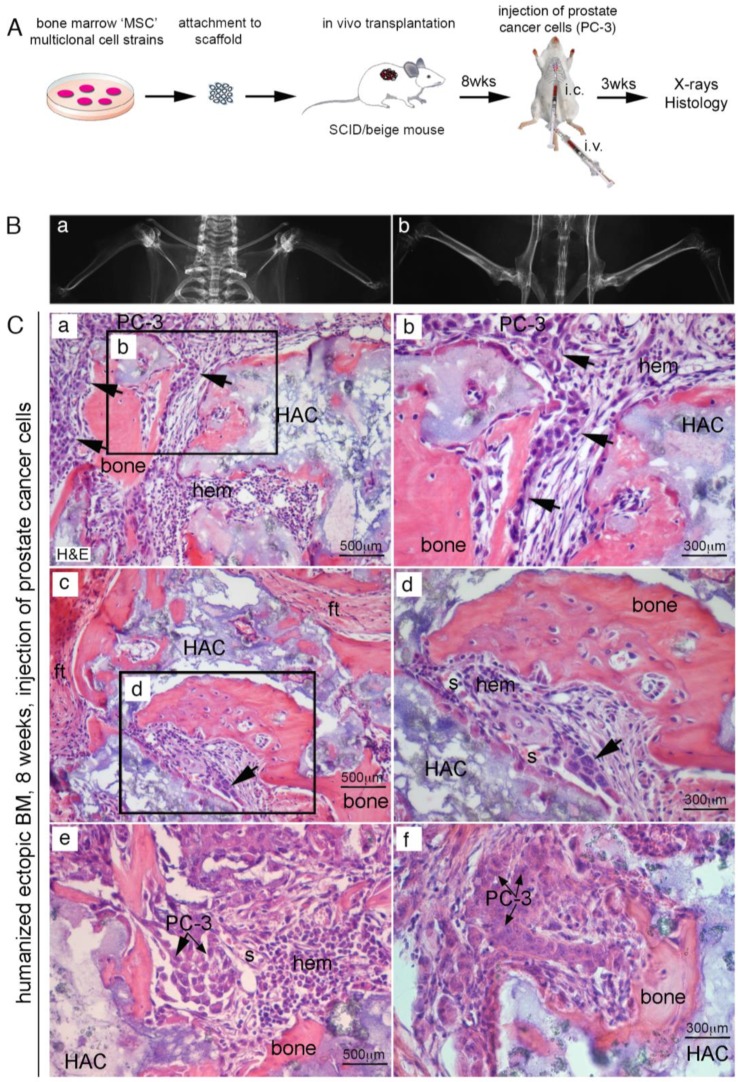
(**A**) experimental design to generate prostate cancer metastasis in humanized bone/BM ossicles. (**A**) PC-3 and RWPE-1 cells were i.c. or i.v. injected in mice bearing heterotopic ossicle; (**B**) radiographic images were not indicative of mice skeleton tumor cell colonization, and histological analyses confirmed the absence of tumor cells in all examined bone tissues; (**C**) only PC-3 cells could generate metastases and only in ectopic humanized bone/BM ossicles (a–f, arrows). H&E, Hematoxylin/Eosin; ft, fibrous tissue; s, sinusoid HA/TCP particles (HAC), hematopoietic cells (hem). Scale bars = 300 μm, 500 μm.

**Figure 5 cancers-11-00763-f005:**
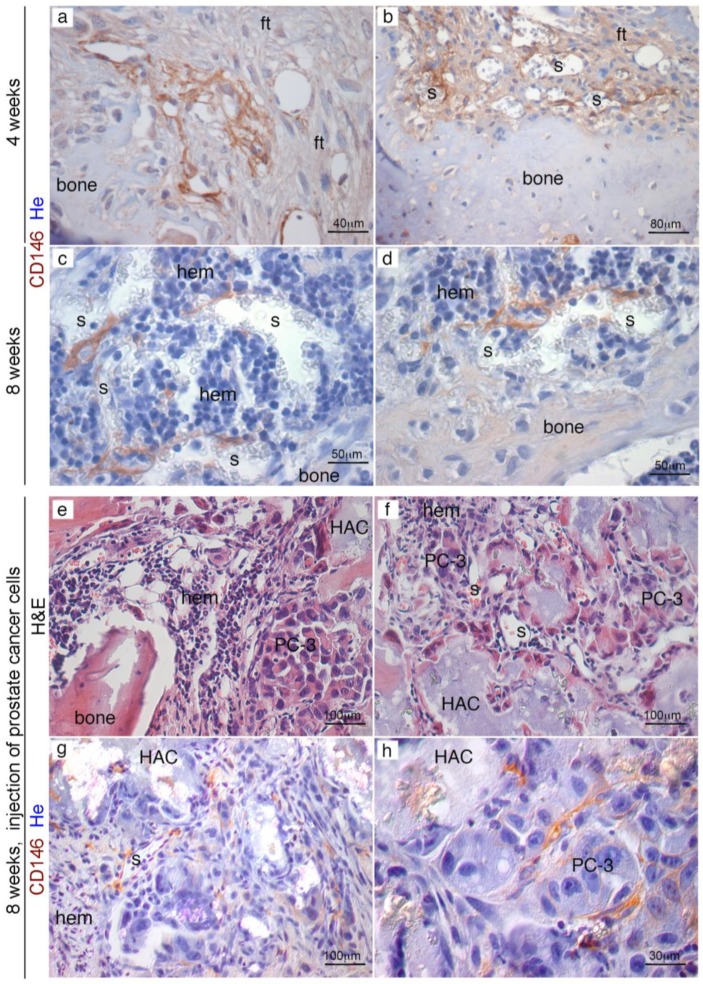
PC-3 cell injection at different times during heterotopic humanized bone/marrow organ development. About four weeks from of hBMSCs transplantation, extraskeletal ossicles present abundant bone and fibrous tissue, bone but not marrow (**a**). CD146^+^ cells are mixed with fibrous cells and are in contact with vessels (**b**). About eight weeks from transplantation, a complete marrow microenvironment has been recreated in extraskeletal ossicle and CD146^+^ can be detected as pericytes around BM sinusoids (**c**,**d**). Injection of PC3 cells four weeks after hBMSCs transplantation resulted in unsuccessful migration/survival of human cancer cells in heterotopic ossicles (**e**,**f**). PC3 cells injected eight weeks from hBMSCs transplants were able to grow in ectopic humanized BM ossicles. Immunolabelling for human CD146 identified both metastatic foci of PC3 cells (**h**) and the human advential reticular cells wrapped around the sinusoid wall (**g**). H&E, Hematoxylin/Eosin; ft, fibrous tissue; s, sinusoid; hem, hematopoiesis, HA, hydroxyapatite. He, Hematoxylin counterstaining. Scale bar = 30, 40, 50, 80, 100 μm.

**Figure 6 cancers-11-00763-f006:**
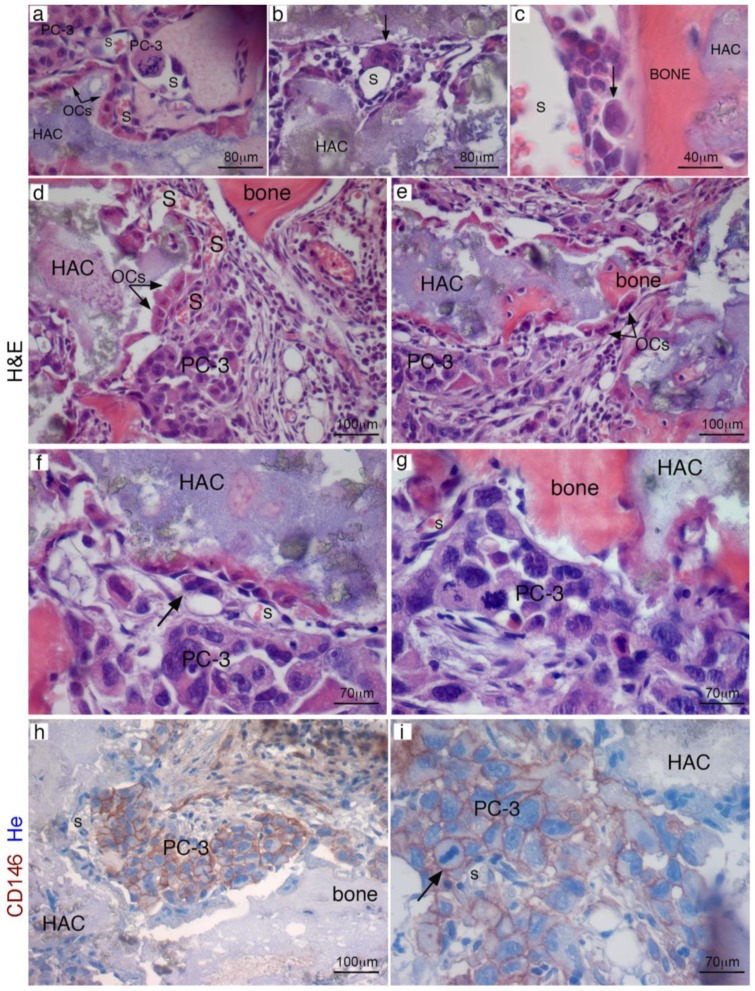
PC3 cells localization in extraskeletal bone/marrow ossicles and foci of cancer cells occupied a perisinusoidal position. In heterotopic transplants, single cancer cells can be observed inside sinusoids (**a**), wrapped around the outer surface of sinusoids (**b**) or near sinusoids, in the space between the vessel and bone (**c**). Early metastatic deposits were regularly associated with sinusoids and with the surfaces of bone or carrier particles (**d**,**e**). Intra-cardiac injected PC3 cells generated massive metastases in transplants (**f**,**g**). PC-3 metastases were identified by immunolabelling for human CD146 in ectopic marrow ossicles (**h**,**i**). HA/TCP particles (HAC), hematopoietic cells (hem); H&E, Hematoxylin/Eosin; OCs, osteoclasts; S, sinusoid. Scale bar = 40, 70, 80, 100 μm.

**Figure 7 cancers-11-00763-f007:**
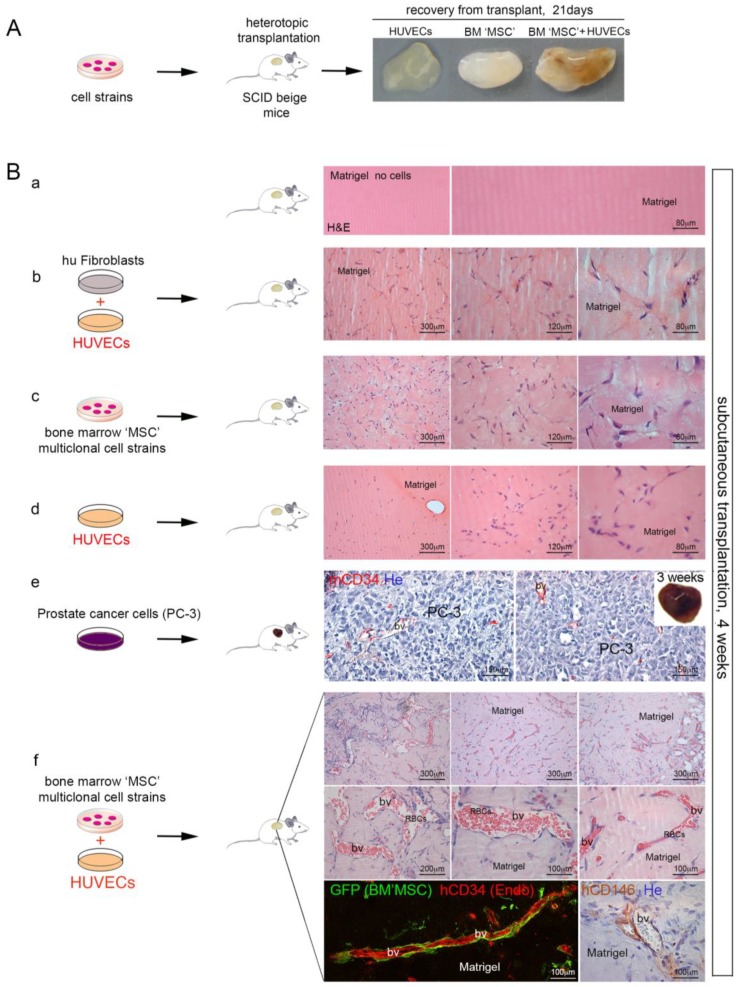
(**A**) formation of vascular networks by BM mesenchymal stem cells (MSCs) *in vivo*. Human foreskin fibroblasts and human BMSCs with or without HUVECs were resuspended in Matrigel^TM^ and implanted on the backs of SCID/beige mice by subcutaneous injection. Gross appearance of ectopic transplants harvested at three weeks. (**B**) H&E staining of implants harvested after four weeks revealed the presence of lumenal structures (bv) containing erythrocytes (RBCs) in implants where both cells HUVECs and hBMSCs cells were used (f) but not in implants where human foreskin fibroblasts were co-transplanted with HUVECs (b), or when human fibroblasts, or hBMSCs (c) were used alone. Images are representative of implants harvested at four weeks. Human PC-3 cells growth efficiently up to form tumors vascularized by murine vessels, when locally grafted in this matrix (e). H&E, Hematoxylin/Eosin; bv, blood vessels; RBCs, red blood cells. He, Hematoxylin counterstaining. Scale bar = 80 μm, 120 μm, 150 μm, 200 μm, 300 μm.

**Figure 8 cancers-11-00763-f008:**
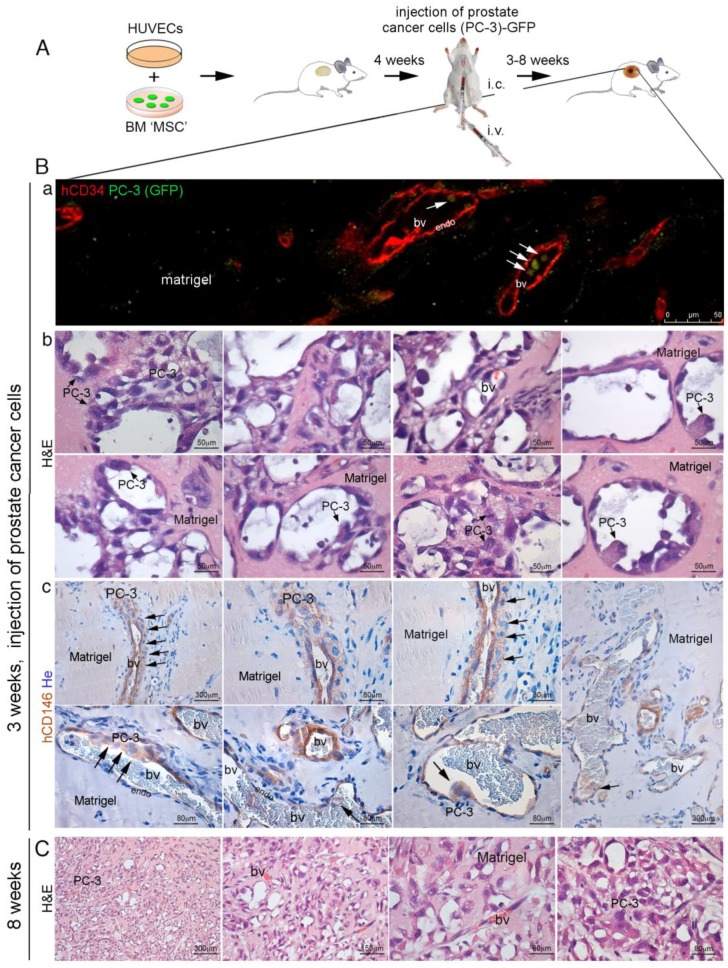
(**A**) in vivo heterotopic co-transplants of hBMSCs in Matrigel^TM^ with HUVECs and injection of PC-3 cells; (**B**) implants generated by hBMSCs and HUVECs in Matrigel^TM^ efficiently formed functional blood vessels in 3–4 weeks. At this time, PC-3 cells were blood-injected in SCID/beige mice. Cancer cells colonized the implants generating metastasis. (a) representative images of implants (hBMSCs/HUVECs in Matrigel^TM^) harvested at 3–4 weeks after injection of PC-3 cells; (b) co-transplants of hBMSCs in Matrigel with HUVECs generated an extensive functional network of capillary-like blood vessels (bv), as indicated by blood host perfusion, in the presence of red blood cells (RBCs). These vessels were composed by two cell layers: an inner continuous hCD34^+^ endothelial layer (endo), coated by an outer subluminal, thin, discontinuous layer of hCD146^+^ mural cells. In many instances, the blood vessels were coated with a multilayer of hCD146^+^ PC-3 cells, wrapped around the outer surface of vessels or near vessels in association with the layer of hCD146^+^ mural cells as demonstrated by hCD146 immunostaining and hematoxylin counterstaining. Early metastatic PC-3 cell clusters were regularly associated with blood vessels. (c) By fluorescence microscopy, single GFP-labeled PC-3 cells could be observed inside blood vessels; (**C**) co-transplants of hBMSCs in Matrigel^TM^ along with HUVECs resulted in the formation of extensive metastases after eight weeks from i.c. injected PC3 cells, as demonstrated by H&E staining. H&E, Hematoxylin/Eosin; bv, blood vessels; RBCs, red blood cells; endo, endothelial cells. He, Hematoxylin counterstaining. Scale bar = 50, 80, 150, 300 μm.

**Table 1 cancers-11-00763-t001:** PC-3 cells selectively home to humanized extraskeletal bone/marrow organs.

Cell Line	Route	Transplants	Transplants with Metastases	Metastases in Mouse Bones
PC-3	i.c.	12	7	-
PC-3	i.v.	8	4	-
RWPE-1	i.c.	8	0	-
RWPE-1	i.v.	12	0	-

**Table 2 cancers-11-00763-t002:** Antibodies used for immunohistochemistry.

Antigen	Type	Cat#	Distributor
human Mitochondria	MM	MAB-1273	Chemicon
human CD146/MCAM	MM	NCL-CD146	Novocastra
human CD34	MM	NCL-END	Novocastra
murine CD34	RaM	ab8158	Abcam
CD3	RP	A0452	Dako
GFP	RP	ab6556	Abcam
GFP	RP	A6455	Invitrogen

MM, Mouse Monoclonal; RaM, Rat Monoclonal; RP, Rabbit Polyclonal.

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
