# Peer review of "Human Sinusoidal Subendothelial Cells Regulate Homing and Invasion of Circulating Metastatic Prostate Cancer Cells to Bone Marrow"

_cancers, 2019, doi:10.3390/cancers11060763_

Round 1
Reviewer 1 Report
The authors describe a mouse models for generation of humanized extraskeletal bone/BM ossicle. Subendothelial cells (pericytes) and self-renewing skeletal stem cells (SSCs) found in bone marrow (BM) stroma when transplanted heterotopically in immunocompromised mice, organize the hematopoietic microenvironment generating a humanized bone/BM ossicle.
They confirmed the hypothesis that bone marrow subendothelial cells committed to a pericyte fate can specify for homing niches for circulating metastatic prostate cancer cells.
Although the article is well written and referenced, the abstract is difficult to read and the title do not directly reflect the content.
Author Response
Author’s Response: We are glad to read the reviewer’s comment. We thank the referee for his/her suggestions. As suggested by the referee, Title and Abstract have been modified to better reflect the content of the manuscript. A review of English was also performed.
Reviewer 2 Report
In the current work, A Funari describes the implication of peri-vascular mesenchymal stem cells to regulate the metastasis to bone marrow. After several readings of the article, this article still remains a puzzle for me and it is difficult to understand and to follow the experimental design. The implicated model seems quite artificial and doubts an the clinical relevance of these findings in human disease.
Major remarks: please include in vitro experiments between CD147 MSC, HUVEC and other MSC to see which cell support PC-3 growth.
Please include also IHC stainings on bone or BM section of patients with bone metastasis to show the implication of the CD147 SSC in the neigbourhood of the prostate cancer cells.
Please join an illustrative figure to explain the experiments and the differences between the experiments, because these are hard to follow.
Author Response
In the current work, A Funari describes the implication of peri-vascular mesenchymal stem cells to regulate the metastasis to bone marrow. After several readings of the article, this article still remains a puzzle for me and it is difficult to understand and to follow the experimental design. The implicated model seems quite artificial and doubts an the clinical relevance of these findings in human disease.
Major remarks: please include in vitro experiments between CD147 MSC, HUVEC and other MSC to see which cell support PC-3 growth.
Author’s Response: We appreciate the reviewer’s comment. Although fundamental, we have not addressed these biological and applicative issues. Indeed:
1) Our goal was to specifically verify the hypothesis that bone marrow sub-endothelial MSC cells committed to a pericyte fate can specify for homing niches for circulating metastatic prostate cancer cells; this to study metastatic bone disease.
2) We wanted specifically to analyze interactions between bone marrow perivascular MSCs and prostate cancer cells in BM microenvironment. For this purpose, we emphasize that human BMSCs were able to generate perivascular cells in vivo, when implanted in ectopic area. To prove this, we developed a new model to answer ongoing questions concerning the epithelial cancer cells – bone marrow MSC stromal interactions in the metastatic process.
3) In this view, we felt reasonably to exclude from our analyses to test any MSCs derived from other tissues.
Furthermore, heterotopic transplants in mice (two mice models developed in our Lab) are more stringent than in vitro studies, and really crucial for investigating the hypothesis that epithelial cancer cells would interact with vascular cell of bone marrow compartment to form metastases.
The overall goal of our paper was to analyze which cells of vascular cell compartment could support PC-3 growth, and to verify if the metastatic processes from cell extravasation to humanized BM niches colonization would depend on species-specific interaction between human BM stromal cells and human cancer cells.
We are not sure that any in vitro possible experiment could replicate these conditions.
Please include also IHC staining on bone or BM section of patients with bone metastasis to show the implication of the CD147 SSC in the neighborhood of the prostate cancer cells.
Author’s Response: We agree with the reviewer’s comment and satisfy her/his request. We have now added an IHC staining on BM section of patients with bone metastasis to show the implication of the CD146 SSC in the neighborhood of the prostate cancer cells (see Supplemental Figure 1) as further proof that these cells can effectively contribute to support prostate cancer cell growth and the clinical relevance of these findings in human disease.
Please join an illustrative figure to explain the experiments and the differences between the experiments, because these are hard to follow.
Author’s Response:
We appreciate the reviewer’s comment. To explain the experiments in the present manuscript, an illustrative Figure 1 reported the experiments design respectively of model using to generate prostate cancer metastasis in SCID/beige mice bearing heterotopic humanized bone/BM ossicles and the experiments design to developed prostate cancer metastasis in SCID/beige mice carrying heterotopic matrix. In addition, each figures, individually, showed the experimental design performed, to better explain, follow and understand, step by step, the experiments of this work.
Furthermore, in our revised manuscript the flow of the experimental design is more clear and detailed, and text more fluent.
Reviewer 3 Report
The manuscript by Funari et al seeks to understand the importance of perivascular cells and their importance in mouse xenograft studies. The authors provide sufficient background in the introduction and a well thought out rationale and initial experiments to show the shortcomings in the current literature. The development of new models to study metastatic bone disease is important, and the authors provide characterization of new models to answer ongoing questions concerning the metastatic process. Overall, the paper is well conceived and written with suitable supporting data concerning the conclusions the authors present.
Minor critique
The authors describe spontaneous murine lymphomas present in their model system. Overall, this is an interesting observation not typically reported. The authors state "a small number of mice" developed these tumors. The reviewer would appreciate slightly more data concerning how many mice were impacted and the incidence of these observations on the total number of mice studied. Citing these figures in the text is sufficient. Known similarities or differences, and comparisons to, previously observed incidences of these spontaneous lymphomas would also be appreciated.
Author Response
Author’s Response: We are flattered by the comment of the reviewer. We thank the referee for his/her suggestions. Statistically, 15% of SCID beige mice spontaneously develop thymic T-cell lymphomas in our work. The same incidences of spontaneous lymphomas in SCID beige mice is also reported in a reference cited in the present manuscript (Bibliography, Reference 12, Custer et al., Am J Pathol). Lymphomas were found in SCID mice of all ages. We totally agree with the reviewer’s comment and fixed the request into the text (line 308 and line 391).